# Mental Health in Schoolchildren in Joint Physical Custody: A Longitudinal Study

**DOI:** 10.3390/children8060473

**Published:** 2021-06-04

**Authors:** Anders Hjern, Stine Kjaer Urhoj, Emma Fransson, Malin Bergström

**Affiliations:** 1Centre for Health Equity Studies, Stockholm University/Karolinska Institutet, Stockholm, 106 91 Stockholm, Sweden; malin.bergstrom@ki.se; 2Department of Medicine, Clinical Epidemiology, Karolinska Institutet, Stockholm, 171 76 Stockholm, Sweden; 3Section of Epidemiology, Department of Public Health, University of Copenhagen, Copenhagen, 1165 København, Denmark; stur@sund.ku.dk; 4Department for Women’s and children’s Health, Uppsala University, Uppsala, 752 36 Uppsala, Sweden; emma.fransson@ki.se

**Keywords:** divorce, parental separation, socioeconomic, social determinants, child, joint physical custody

## Abstract

This study investigated mental health in schoolchildren in different living arrangements after parental separation. The study population included 31,519 children from the Danish National Birth Cohort, followed-up at age 11 in 2010–2014. Child mental health was measured with a maternal report of the Strength and Difficulties Questionnaire (SDQ). Associations between living arrangements and mental health were analyzed using logistic and linear regression models, taking into account early childhood indicators of the parents’ relations, income, education and psychiatric care. At age 11, children living in a nuclear family had the lowest rate of total SDQ score, 8.9%. Of the children who had experienced parental separation, children in joint physical custody had the lowest adjusted odds ratio (OR)1.25 (95%-CI 1.09–1.44), for a high SDQ score relative to children living in a nuclear family, with adjusted ORs of 1.63 (1.42–1.86) and OR 1.72 (1.52–1.95) for sole physical custody arrangements with and without a new partner. An analysis of change in SDQ scores between ages 7 and 11 in children showed a similar pattern. This study indicates that joint physical custody is associated with slightly more favorable mental health in schoolchildren after parental separation than sole physical custody arrangements.

## 1. Introduction

Parental separation has become a life experience shared by many children in high-income countries. In Scandinavia, more than one in three children experience parental separation before reaching adulthood [1]. In the literature, it is well-established that children with separated parents suffer from elevated risks for ill health and problems regarding their cognitive, social, emotional and behavioral development [2]. These risks may be related to the pre-separation characteristics of children as well as of the family, to the crisis brought along by the separation [3] or children’s life circumstances after the separation [4]. Losing contact with one parent, most often the father, has been described as especially difficult for children [5]. In recent decades, however, parenting norms have changed, with fathers becoming more involved after the separation, in particular in the Nordic countries [6]. As a consequence, joint physical custody has been established as a common arrangement after parental separation. Joint physical custody refers to a practice where children with non-cohabiting parents live alternating between and essentially equally as much with both parents. The share of joint physical custody among children with separated parents has been reported to be 25% in Denmark and Norway and about 35% in Sweden [1]. Studies on how children fare in different family types after the separation may inform policy and family law about factors that contribute to children’s health in families with separated parents. However, keeping everyday contact with a father after a parental separation may also have a cost for children.

Joint physical custody (JPC) has been suggested to imply health risks, such as the stress of living in two homes and in two different family cultures [7,8], and difficulties in maintaining social contacts when moving between two neighborhoods [9]. However, recent literature reviews find more positive child outcomes in joint physical custody compared to sole physical custody arrangements [10]. Spending substantial time with both parents is beneficial for the parents–child relationship. These advantages in well-being for children may, however, be related to characteristics of families who choose this family type (6). Studies have consistently shown that parents in joint physical custody families have higher educational levels, more often dual incomes and positive co-parenting relationships than those where the children reside only with the mother [11]. This positive selection into joint physical custody poses an important analytic challenge in comparative studies of living arrangements after parental separation. Furthermore, the literature consists almost entirely of cross-sectional studies with little or no information about family factors preceding the separation [12,13,14]. The risk of bias associated with this study design and the need for longitudinal studies with information on such factors were mentioned already in early studies on this topic [15,16].

In a recent study in the Danish National Birth Cohort, we found that socioeconomic indicators, parental relations and mental health in early childhood were important predictors of parental separation as well as the living arrangements of the child after divorce at age 11. Having a more favorable socioeconomic situation and less-strained family relations increased the odds for joint physical custody after parental separation compared with the odds of living in a sole physical custody arrangement. These results, together with the well-established associations of mental health in schoolchildren with socioeconomic determinants [17] and parental risk factors [18], indicate that the mental health outcomes found in earlier studies of joint physical custody have been confounded by pre-separation factors. These results were confirmed by a recent large-scale study on children in 37 countries, where the advantages of children in joint physical custody became statistically nonsignificant when the analyses were adjusted for family affluence and parent–child relationship quality [19].

By making the best use of the longitudinal design of the Danish National Birth Cohort, the aim of this study was to investigate mental health in children in joint physical custody compared with other living arrangements at age 11, taking into account risk factors preceding parental separation.

## 2. Materials and Methods

This study builds on data collected in the Danish National Birth Cohort linked to Danish national registers. The study design and response rates of this cohort have been described elsewhere [20]. Women from all regions of Denmark were invited at their first pregnancy visit to their general practitioner. In total, approximately 60 percent of all invited pregnancies were recruited; 92,274 women and their 100,415 pregnancies [20]. Data collections started during pregnancy, but this study used data from computer-assisted telephone interviews with the mother when the child was 6 months and 7 and 11 years old. The child was interviewed at age 11, and this interview was used for the categorizing of living arrangements at that age.

In total, 35,301 children who lived with both parents at the age of 6 months provided information about their current living arrangements in the data collection at 11 years, corresponding to 51.6 percent of the children invited to the 11-year follow-up. Questionnaire data from the mother and information from the national registers provided data for 31,519 children that fulfilled the living arrangement criteria of this study. These children constituted the overall study population. Of these, 24,084 children had maternal-reported mental health data at the 7-year interview. The selection process and reasons for exclusion are presented in Figure 1.

Baseline characteristics of study participants and dropouts were compared in an attrition analysis, presented in Appendix A. More details on the study design and questionnaires are available at dnbc.dk and in past publications describing [20] and using [21,22] this cohort data. 

### 2.1. Living Arrangements

The categorization of living arrangements at the 11-year follow-up was based on one survey item in the child questionnaire asking which adults the child lived with all or most of the time. This item had nine possible response categories that were merged into four categories in this study: a. nuclear family (“with both my parents”); b. joint physical custody (“I split my time equally/almost equally between my mother and my father”); c. sole physical custody without a new partner (“with my mother/with my father”); d. sole physical custody with a new partner (“with my mother and her new boyfriend/husband; with my father and his new girlfriend/wife”). The 228 children who did not fulfill the criteria for any of these categories (“foster family; live in an institution; other”) were excluded from the study population (Figure 1).

### 2.2. Mental Health Outcomes

Mental health was measured with the validated Danish parent-report versions of the Strengths and Difficulties Questionnaire (SDQ) [23,24,25]. The SDQ scores were operationalized as two outcomes. First, a high total SDQ score at age 11 for the entire study population, defined as a score of 12 or more, in accordance with the Danish validity study of SDQ [24]. Second, as the mean difference (MD) in total SDQ scores from age 7 to 11. 

### 2.3. Early Childhood Indicators

Two items from the 6-months interview with the mother that were previously found to predict both parental separation and living arrangements in the Danish National Birth Cohort were included in the study [26]. These two items asked (i) whether the relation to the father was a burden and (ii) whether the economic situation of the family was a burden, with three possible responses: no, some, yes. 

In addition to the information provided by mothers and children in the cohort, the study also included covariates based on register data from Statistics Denmark linked through the unique person identifier existing in these registers. The age and sex of the child and the maternal age at the birth of the child were obtained from the Danish Medical Birth Register [27]. Information on the disposable income of the family was retrieved from the Income Statistics Register [28] and information on parental education from the Population Education Register [29]. Disposable household income was equivalized by use of the OECD-modified equivalence scale and divided into quintiles by year of birth and parental education represents the highest level attained. Data on any contact with secondary mental health services before the birth of the child were retrieved from the Danish Psychiatric Central Research Register [30] from 1977 onwards and was dichotomized as yes or no. 

### 2.4. Analysis

The analytic strategy of investigating mental health in the different living arrangements consisted of two steps. In the first step, a high SDQ difficulties score at age 11 was analyzed in a logistic regression with adjustment for early childhood predictors of living arrangements after parental separation found in a previous study of the Danish National Birth Cohort [26]. These factors were found to also predict high SDQ difficulties scores in this study and were thus considered as confounders in the statistical analysis. Odds ratios (ORs) and 95% confidence intervals (CIs) were calculated in three logistic regression models. In Model 1, only gender was adjusted for. In Model 2, indicators of parental psychiatric disorders before the birth of the child and a dichotomized indicator of the mother reporting the relation to the father to be a burden (some/much vs. no) from the 6-months’ interview were added. In the fully adjusted Model 3, disposable household income in quintiles, maternal and paternal education prior to the birth of the child, mother considering the economy a burden at the 6-months’ interview and maternal age were added to the variables in Model 2.

In the second set of analyses, we used data from the 24,749 children whose mothers participated in both the 7-year and the 11-year interviews and fulfilled the criteria for living arrangements at both interviews. A variable of mean difference (MD) of total scores of SDQ was calculated by subtracting mean total scores at age 7 from that of total scores at age 11. This variable was found to have a Gaussian distribution (See Appendix A). This outcome variable was analyzed in two separate analyses. The first analysis included children who had experienced parental separation before age 7 and lived in the same living arrangement (joint physical custody, sole physical custody with new partner, sole physical custody with no new partner) in both data collections. The second analysis included those who were living in a nuclear family at age 7 but had experienced parental separation between the ages of 7–11. Children living in a nuclear family at both time points were included as a reference population in both analyses. Beta-values and 95% confidence intervals (CI) were calculated in a linear regression with the adjustments of Model 1 and 3 above.

Gender differences in the investigated associations were examined by assessing interaction with gender in all models, but since no statistically significant interactions on the multiplicative scale were found, all results are presented without gender stratification. Analyses were made using SPSS 27.0 (IBM, Chicago, IL, USA).

## 3. Results

At the 11-year follow-up, 31,519 children in the 68,427 families with cohabiting parents who participated in the 6-months data collection were included in the population for this study (see Figure 1). In an attrition analysis (Appendix A), we found that mothers lost to follow-up more often had a history of psychiatric care (7.4% versus 4.9% in participating mothers), more often were mothers of sons (52.8% versus 48.4%), were slightly younger and less often had a university degree. In the study population, 79.9% of the children were living in a nuclear family, 6.9% in joint physical custody, 6.7% in a sole physical custody arrangement without a new partner and 5.6% in a sole physical custody arrangement with a new partner. 

### 3.1. Descriptive Analyses

Table 1 shows the sociodemographic, parental psychiatric disorder and parental relational covariates in early childhood by child-reported family arrangements at the 11-year follow-up. Children living in a nuclear family at 11 years more often had parents with higher incomes and a high educational level and less often had a psychiatric disorder, compared with children living in a sole physical custody arrangement. Parents of children living in joint physical custody had an educational and income level similar to that of children living in a nuclear family but had moderately higher rates of parental psychiatric disorders before the birth of the child.

The proportion of children with a high total SDQ score at 11 years was 10.2% in the total study population. This figure was 8.9% for children living in a nuclear family, 11.7% for children in joint physical custody and 17.9% for children without a new parental partner and 18.2% for children living in sole physical custody. The proportion of children with a high SDQ score varied by gender: 11.7% of boys and 8.9% of girls. The proportion also varied gradually from low to high by all SES indicators, with the gradient being particularly pronounced for maternal and paternal education (Table 2).

In the analysis of change in mean difference (MD) in total SDQ scores between ages 7 and 11, the overall beta was −0.04 (95% C.I. −0.10–0.02) with a standard deviation of 3.7. In children living in a nuclear family at both data collections, the total SDQ scores decreased slightly over time (Beta of MD: −0.09), while the MD in the children who had experienced parental separation varied from MD 0.27 in children in joint physical custody to MD 0.63 in sole physical custody without a new partner. 

### 3.2. Logistic Regression Analysis

Table 3 shows the results from the logistic regression model with a high total SDQ score at the 11-year interview. In comparison with children in a nuclear family, children in joint physical custody had an OR of 1.25 (95%-CI 1.09–1.44) in the fully adjusted Model 3; children living in a step-family and with sole physical custody with no new partner had ORs of 1.63 (95%-CI 1.42–1.86) and 1.72 (95%-CI 1.52–1.95), respectively. For children living in sole physical custody arrangements, ORs were much attenuated when confounders were added in Models 2 and 3, while this attenuation was more moderate for children in joint physical custody.

### 3.3. Analysis of Change between Age 7 and Age 11 

Of the children who were included in the study population at age 11, 24,749 also had maternal reports of SDQ at age 7. Of these children, 20,137 lived with both parents at the time of both interviews. Of the remaining 3947 children, 1804 children had the same living arrangement at age 7 and age 11, while for 1531, the living arrangement changed between ages 7 and 11. In an analysis of change in mean total SDQ scores, between ages 7 and 11, these two groups of children were analyzed separately with the children living in a nuclear family as the reference population; see Figure 2 and Figure 3 and Table 4. 

Figure 2 illustrates the change in total SDQ scores from age 7 to age 11 by living arrangements in children who experienced parental separation before age 7 and remained in the same living arrangement at age 11. Figure 3 illustrates the same in children who experienced parental separation between ages 7 and 11. Children who lived with both parents at ages 7 and 11 are included in both figures as points of reference. As expected, the changes in SDQ scores were greater in the children who more recently had experienced parental separation.

Table 4 shows the results from a linear regression analysis of the mean difference (MD) in SDQ scores between ages 7 and 11. For children who had experienced parental separation before age 7, those having lived in joint physical custody had a similar adjusted *beta* to those living in a nuclear family, −0,21 (95% CI −0.63–0.20), while those living in sole physical custody arrangement with no new partner had the highest adjusted estimate of change, beta 0.38 (0.10–0.65). 

For those who had experienced parental separation between ages 7 and 11, children in joint physical custody again had the lowest adjusted beta, 0.47 (0.21–0.73), compared to children living in a nuclear family. Children living in a sole physical custody arrangement with a new partner had the highest adjusted beta, 0.85 (0.56–1.14).

## 4. Discussion

In this study of mental health in more than 31,500 children from the Danish National Birth Cohort, we found that 11-year-old children with separated parents had more mental health problems than children living with both of their parents together. Mental health also varied between children in different family types after parental separation. Children in joint physical custody had slightly better mental health outcomes compared with children in sole physical custody with no new partner and step-family arrangements after parental separation, before and after adjustment for pre-separation risk factors. Longitudinal analyses showed that among children with stable family types between ages 7 and 11, SDQ mean scores were slightly increased for children in sole physical custody arrangements but decreasing for those in joint physical custody and nuclear families. For children who experienced a parental separation between ages 7 and 11, mean SDQ scores increased, but to a lower extent for children in joint physical custody compared with children in a sole physical custody arrangement. 

There are several theoretical perspectives in the literature to explain why children suffer from parental separation [31]. For young schoolchildren, as those included here, parents still play a crucial role in the children’s development of emotional and behavioral regulation [32]. During the actual separation crisis, the parents’ ability to support these processes is naturally impeded. Their ability to reestablish familial security and stability post-separation is thus crucial for children’s recovery and subsequent mental health [33]. 

Our results indicate that parents in joint physical custody families manage to establish such conditions, despite the obvious disadvantages for children in terms of two different home environments and frequent moves (including related logistic problems) between the homes [34] (RAKEL). A plausible explanation for these children’s positive outcomes may be a more positive relationship between the children and their parents, and in particular to their fathers, which joint physical custody enables [35,36,37]. Swedish data show that children in joint physical custody are as satisfied with their parental relationships as children in nuclear families [38]. Thus, living together with one’s parents appears to strengthen relationships and may possibly contribute to better mental health in children in joint physical custody. For children in sole physical custody with no new partner arrangements, the parental separation instead entails a break, not only in the relationship between the parents, but also in children’s contact with the parent with whom they cease to live together. This loss of closeness to a parent (most often the father) is associated with negative consequences for children’s mental health as well as for their self-esteem [39,40]. Sole physical custody with no new partners, who more often than parents in two-parent and joint physical custody families have low educational levels and incomes, also face a parenting situation where they alone must provide both emotional parenting support, set boundaries and support the child financially. The stress associated with these demands has been shown to increase the risks for low satisfaction with life and increased health risks (for sole physical custody with no new partners). A step-parent could potentially contribute to lessening such stress by becoming a secure parenting figure and by providing economic resources. However, our results, like earlier publications, do not support this hypothesis [41]. A potential explanation for children’s mental health problems in step-families is that the introduction of a step-parent in the household may hamper the family stability after the parental separation. For children in step-families, the parental break-up is the first of several family transitions [42]. Parallel to the child’s loss of everyday contact with one parent after the parental separation, the relationship with the custody parent may also be influenced by the new partner and possibly also by the presence of step- and half-siblings [43]. How step-parents negotiate their parental role and establish their co-parenting norms also influences child well-being [44]. Additionally, step-parents may not bring about resources to the child as would be expected, since parents tend to invest less in children who are not biologically related to them [45]. Since step-parents often introduce half- and step-siblings, their economic contribution may in fact be negative [44]. 

It is, however, important to note that the difference in mean difference in SDQ between joint physical custody and sole physical custody with no new partner arrangement is only about 10% of a standard deviation for this measure and must thus be considered very small. A favorable pattern for schoolchildren in joint physical custody after parental separation has previously been repeatedly shown in population-based cross-sectional studies with numerous outcomes such as satisfaction with life, risk behavior, school achievement, well-being and mental health [1,13,14,15,46]. In this first large longitudinal study on this topic, we have accounted for multiple early childhood predictors of child mental health in the logistic regression, which attenuated differences between children living in a nuclear family and sole physical custody arrangements, while attenuation was more moderate for children in joint physical custody. In the linear regression of change in SDQ score after parental separation between ages 7 and 11, we have furthermore accounted for the combined effect of unmeasured confounders varying between the groups that do not change over the course of the study period, such as genes. The results indicate that the findings in the previous cross-sectional studies were not artifacts. However, the considerable attenuation of the difference between joint physical custody and sole physical custody arrangements in mental health after adjustment for known confounders and the comparatively small differences in effects in the analysis of change over time suggests that previous studies may have overestimated the advantage of joint physical custody. 

In a previous study, in the Danish National Birth Cohort, no mental health advantage of joint physical custody relative to sole physical custody without a new partner was found in children with a separated parental age of 7 years [47]. Thus, the slight advantage of joint physical custody at age 11 found in this study should not be extrapolated to younger ages.

This study, as many previous studies, shows that social determinants like education, low income and young parental age are robust predictors for parental separation [48,49]. This study further suggests that a child living arrangements after parental separation may modify these associations with joint physical custody, leading to slightly more beneficial child outcomes. However, joint physical custody is clearly associated with a more favorable situation with regard to parental education and income. To some extent, this is probably a consequence of the increased costs involved in providing two homes for the child, preventing some parents with low incomes to choose this living arrangement [35]. Hence, joint physical custody potentially adds another layer to the social determinants of child health by reducing the negative consequences of parental separation more in children from privileged social circumstances. 

The main strength of this study is the longitudinal design in a large sample of children who experience parental separation during the course of the data collection, enabled by the large Danish National Birth Cohort. Another strength is the categorization of joint physical custody. Here, children live half or approximately half the time with each parent, while other studies include children spending only 30% of the time with one parent^5^. Finally, the addition of objective measures of SES and psychiatric disorders from national registers provided high-quality adjustment for such confounders. 

The main limitation is the large and complex attrition. The attrition analysis (Appendix A) indicates that children with separated parents and children in families with a low socioeconomic position more often were lost to follow-up. The association of low socioeconomic status with both a high SDQ score and not living in joint physical custody after parental separation could potentially have biased our analyses towards an underestimation of the difference between joint physical custody and sole physical custody with no new partner arrangement after divorce. 

Another limitation of this study is the use of maternal reports as the sole source of information about child mental health. Further studies with child and teacher informants are needed to confirm the findings of this study. Finally, the variables related to family relations describe the situation at 6 months after the birth of the child only and lack important aspects such as domestic violence. 

In conclusion, this study indicates that after parental separation, joint physical custody is slightly more favorable for child mental health in schoolchildren than a sole physical custody arrangement.

## Figures and Tables

**Figure 1 children-08-00473-f001:**
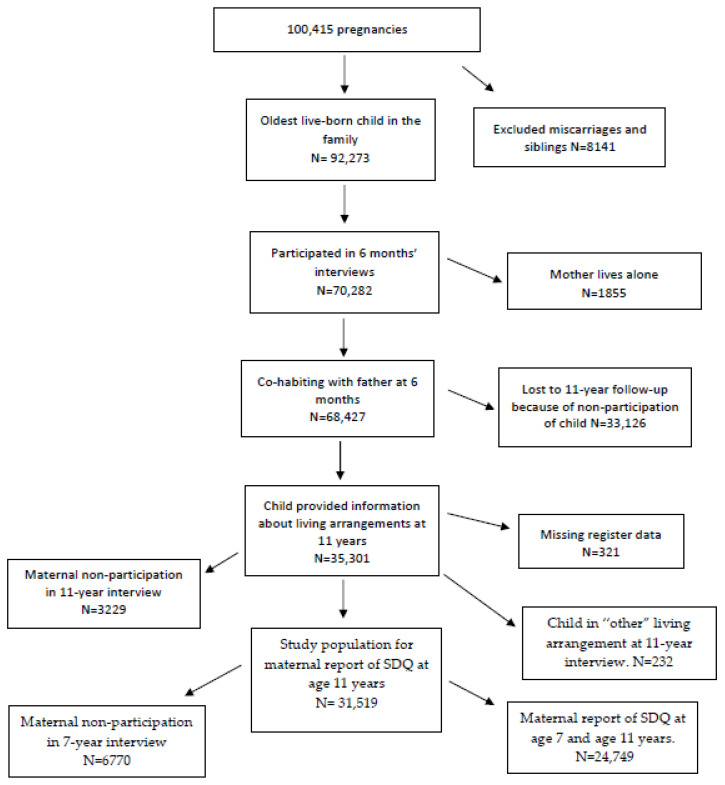
Participation at different data collection points of the study. (SDQ = the Strengths and Difficulties Questionnaire).

**Figure 2 children-08-00473-f002:**
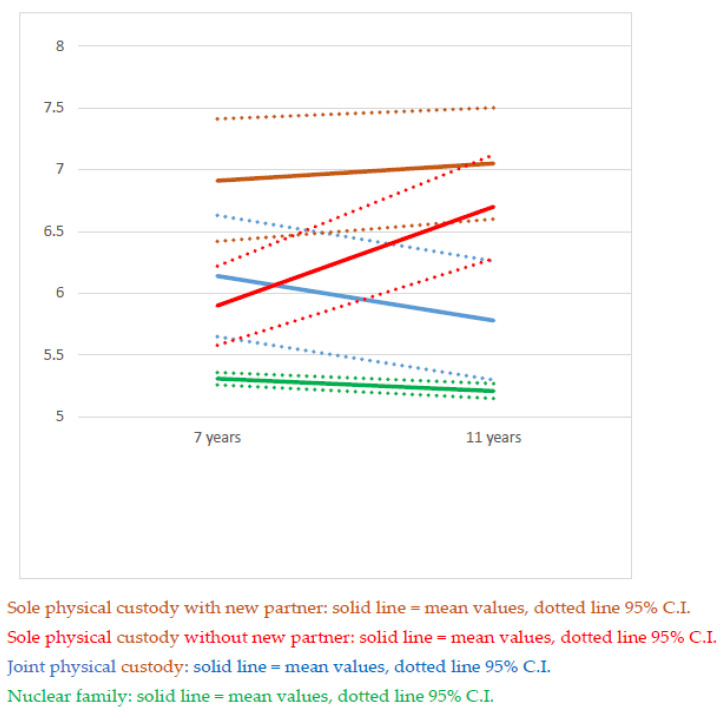
Total SDQ scores with 95% CI in children who were in the same living arrangement at age 7 and age 11 by age at data collection and living arrangement.

**Figure 3 children-08-00473-f003:**
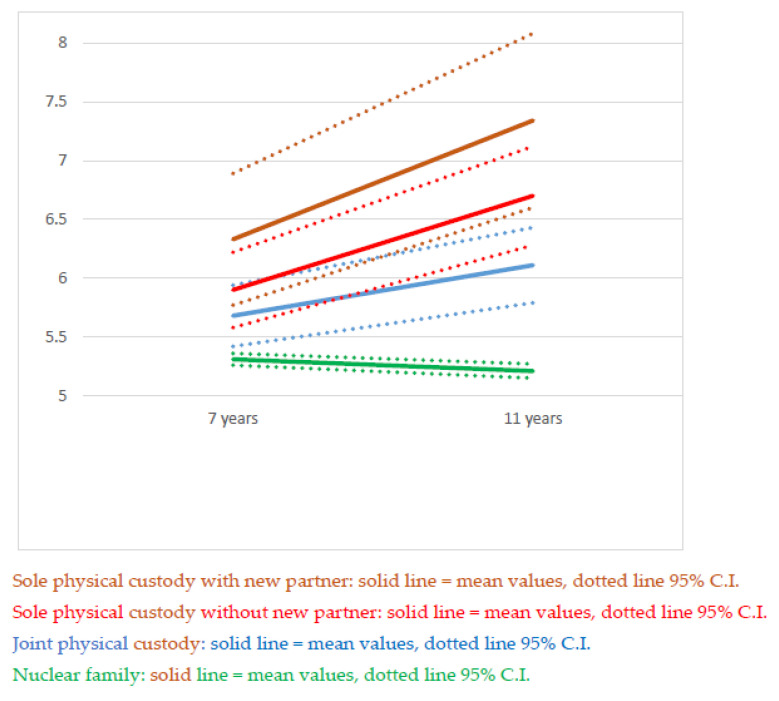
Total SDQ scores with 95% CI in children who lived with both parents at age 7 by age at data collection and living arrangement.

**Table 1 children-08-00473-t001:** Characteristics of the study population by organization of living arrangements at the 11-year interview (*n* = 31,519).

	Nuclear Family	Joint Physical Custody	Sole Physical Custody with New Partner	Sole Physical Custody without New Partner
	*n* = 25,468	*n* = 2188	*n* = 1779	*n* = 2084
	%	%	%	%
Gender				
Boys	49.0	47.5	44.6	46.4
Girls	51.0	52.5	55.4	53.6
Maternal age at birth of child				
15–22	1.2	3.3	6.9	3.1
23–28	32.2	36.9	44.9	31.5
29–34	49.9	46.0	39.5	44.6
35+	16.7	13.8	8.7	20.8
Disposable income before birth				
Quintile 1	14.4	18.3	25.2	22.1
Quintile 2	18.9	18.6	23.5	22.6
Quintile 3	20.8	20.1	21.8	21.5
Quintile 4	22.6	21.3	17.3	18.8
Quintile 5	23.3	21.7	12.1	15.0
Maternal education before birth of child				
Primary	4.2	4.3	8.4	8.6
Secondary	34.8	33.5	47.0	42.0
1–3 years post-secondary	44.4	43.8	35.6	38.9
4 + years post-secondary	16.6	18.4	8.9	10.5
Paternal education before birth of child				
Primary only	8.9	8.4	21.0	17.2
Secondary	44.0	42.7	54.5	50.9
1–3 years post-secondary	28.7	30.5	17.7	21.1
4 + years post-secondary	18.5	18.4	6.8	10.8
Parental psychiatric disorder before birth of child				
Father	3.6	5.3	10.4	11.5
Mother	4.2	6.2	8.2	7.8
Mother burdened by relation to father at child age of 6 months				
No	90.3	80.3	78.4	75.1
Some or much	9.7	19.7	21.6	24.9
Mother burdened by economy at child age of 6 months				
No	83.8	79.0	73.6	73.6
Some or much	16.2	21.0	26.4	26.4

**Table 2 children-08-00473-t002:** Percentage of high SDQ score at 11 years and the mean difference (MD) in SDQ scores from age 7 to age 11 by living arrangements and covariates.

	High SDQ at 11 yrs		MD in SDQ Scores 7 yrs to 11 yrs	
	*n*	%	*n*	Beta
Living arrangements at age 11				
Nuclear family	25,468	8.9	20,285	−0.09
Joint physical custody	2188	11.7	1607	0.27
Sole physical custody with a new partner	1779	18.2	1296	0.32
Sole physical custody without a new partner	2084	17.9	1561	0.63
Gender				
Boys	15,286	11.7	12,115	−0.04
Girls	16,233	8.9	12,634	0.01
Maternal age at birth of child				
15–22	577	23.6	426	−0.22
23–28	10,454	11.1	8125	−0.21
29–34	15,339	9.5	12,049	0.01
35+	5149	9.2	4149	0.36
Disposable income before birth				
Quintile 1	4969	12.6	3770	0.08
Quintile 2	6118	11.4	4821	0.06
Quintile 3	6572	10.7	5206	0.06
Quintile 4	6931	9.7	5484	−0.04
Quintile 5	6929	7.7	5468	−0.14
Maternal education before birth of child				
Primary	1498	18.0	1140	0.33
Secondary	11,311	13.1	8764	−0.01
1–3 years post secondary	13,714	8.3	10,901	−0.03
4 + years post-secondary	4996	6.9	3949	−0.04
Paternal education before birth of child				
Primary only	3177	16.4	2483	0.14
Secondary	14,164	11.7	11,064	0.03
1–3 years post-secondary	8724	7.8	6914	−0.04
4 + years post-secondary	5454	6.7	4288	−0.12
Parental psychiatric disorder before birth of child				
Father	1466	14.5	1121	0.13
Mother	1521	16.2	1151	0.35
Mother burdened by relation to father at 6-month interview				
Some or much	3803	15.9	2498	0.27
Mother burdened by economy at 6-month interview				
Some or much	5599	15.9	4298	0.20
All	31,513	10.2	24,749	−0.04

**Table 3 children-08-00473-t003:** Logistic regression analysis of high SDQ score at age 11 by living arrangements (*n* = 31,519).

		Model 1 ^1^	Model 2 ^2^	Model 3 ^3^
	*n*	OR (95% CI)	OR (95% CI)	OR (95% CI)
Nuclear family	25,468	1	1	1
Joint physical custody	2188	1.36 (1.19–1.56)	1.27 (1.10–1.46)	1.25 (1.09–1.44)
Sole physical custody with new partner	1779	2.30 (2.03–2.62)	2.10 (1.84–2.39)	1.63 (1.42–1.86)
Sole physical custody without new partner	2084	2.26 (2.00–2.55)	2.02 (1.79–2.28)	1.72 (1.52–1.95)

^1^ Model 1 is adjusted for child gender only; ^2^ Model 2 is adjusted for child gender, parental psychiatric conditions before the birth of the child and maternal satisfaction with father at the child’s age of 6 months; ^3^ Model 3 is adjusted for child gender, parental psychiatric conditions before the birth of the child, maternal satisfaction with father at the child’s age of 6 months and early childhood indicators of sociodemographic covariates.

**Table 4 children-08-00473-t004:** Linear regression analysis of change in mean total SDQ score between ages 7 and 11 by living arrangements.

	Same Living Arrangement at 7 yrs and 11 yrs				Lived in Nuclear Family at Age 7	
		Model 1 ^1^	Model 2 ^2^		Model 1 ^1^	Model 2 ^2^
Living arrangement	*n*	Beta (95% C.I.)	Beta (95% C.I.)	*n*	Beta (95% C.I.)	Beta (95% C.I.)
Sole physical custody without new partner	804	0.45 (0.18–0.73)	0.38 (0.10–0.65)	739	0.91 (0.62–1.20)	0.85 (0.56–1.14)
Sole physical custody with new partner	383	0.17 (−0.22–0.56)	0.10 (−0.30–0.49)	286	1.02 (0.56–1.48)	0.97(0.51–1.43)
Joint physical custody	344	−0.19 (−0.61–0.23)	−0.21 (−0.63–0.20)	933	0.50 (0.24–0.76)	0.47 (0.21–0.73)
Nuclear family	20,137	1	1	20,137	1	1

^1^ Model 1 is adjusted for child gender only; ^2^ Model 2 is adjusted for child gender, parental psychiatric conditions before the birth of the child, maternal satisfaction with father at the child’s age of 6 months, and early childhood indicators of sociodemographic covariates.

## Data Availability

Information about access to the data from the Danish National Birth cohort can be obtained at https://www.dnbc.dk/access-to-dnbc-data (3 June 2021).

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
