# Peer review of "Mental Health in Schoolchildren in Joint Physical Custody: A Longitudinal Study"

_children, 2021, doi:10.3390/children8060473_

Round 1
Reviewer 1 Report
Thank you for the opportunity to read this quite interesting paper. This is quite a robust research project that offers readers the opportunity to contemplate these crucial post separation issues from a longitudinal perspective.
A challenge for statistically based research papers is one of accessibility for larger audiences as opposed to fellow researchers. I think this is a difficult balance to achieve. As presently written, the paper is a challenge for the practitioner audience whom I think might well benefit from the results. Perhaps this journal is not going to reach that audience as it will mainly be read by researchers and academics. You have correctly noted that there is a paucity of research in this area. Given that, I wonder about a paragraph that offers a broader audience the take away from this research.
The work is quite sound and I am not criticizing the writing. It is simply that too often this material gets lost in academia and fails to influence families, lawyers and courts.
The paper needs some minor editing for grammar and spelling.
For clarity, the role of the mothers as reporters and the role of the child as reporter of data for the paper needs a few sentences. It gets a bit confusing as to what information the child gave and what the mothers gave.
On the version I received, Figures 2 and 3 had incomplete legends in that the solid versus dotted lines was not explained. I downloaded twice and that was not resolved. This may simply be a formatting error.
Overall, nice work.
Author Response
Dear reviewer.
Thank you for the appreciation of the robustness of our research!
- We certainly agree with the need for this research to be made accessible for practitioners and legal professionals that do not easily follow our scientific style of writing. However, scientific writing has it's own demands of preciseness and nuance that stands in the way. Our way of making our work more accessible has been through media (Ms Bergström is a columnist in the most respected Swedish daily Dagens Nyheter) and to write special articles that summarize findings from several articles in a more accessible; see for instance "What Can We Say Regarding Shared Parenting Arrangements for Swedish Children?" in the Journal of Divorce and Remarriage in 2018.
- We have made some minor language corrections.
- Which data was reported by the mother and which data was reported by by child has been rewritten in the methods section:
"Data collections started during pregnancy, but this study use data from computer-assisted telephone interviews with the mother when the child was six months, 7 and 11 years. The child was interviewed at age 11 years, and this interview was used for the categorizing of living arrangements at that age."
- The legends to Figures 2 and 3 have been revised to offer more clarity.
Reviewer 2 Report
- This is an interesting study
However,
- Some “grammatical” errors should be corrected.
Examples:
- Please change “In the literature it is well established” to “In the literature, it is well established”
- Please change “Nuclear familiy” to “Nuclear family” in the Tables (more than one time!)
- The information related to the parents is not “complete”. As an example, parental education and parental psychiatric disorder, included in the Tables, are important characteristics. However, the history of domestic violence could dramatically influence the sole physical custody (with the “good” or the “bad” parent).
Author Response
Dear reviewer.
Thank you for finding our work interesting!
- The suggested changes of language and misspelled labels have been corrected, as have some other language mistakes.
- The point about the crudeness of our family relations variables is well taken. The following text has been added to the limitation section of the discussion:
"Finally, the variables related to family relations describe the situation six months after the birth of the child only and lack important aspects, such as domestic violence."